# EFFECTIVE SUBSPACE INDEXING VIA INTERPOLATION ON STIEFEL AND GRASSMANN MANIFOLDS

## ABSTRACT

We propose a novel local Subspace Indexing Model with Interpolation (`SIM-I`) for low-dimensional embedding of image data sets. Our `SIM-I` is constructed via two steps: in the first step we build a piece-wise linear affinity-aware subspace model under a given partition of the data set; in the second step we interpolate between several adjacent linear subspace models constructed previously using the "center of mass" calculation on Stiefel and Grassmann manifolds. The resulting subspace indexing model built by `SIM-I` is a globally non-linear low-dimensional embedding of the original data set. Furthermore, the interpolation step produces a "smoothed" version of the piece-wise linear embedding mapping constructed in the first step, and can be viewed as a regularization procedure. We provide experimental results validating the effectiveness of `SIM-I`, that improves PCA recovery for SIFT data set and nearest-neighbor classification success rates for MNIST and CIFAR-10 data sets.

## 1 INTRODUCTION

Subspace selection algorithms have been successful in many application problems related to dimension reduction (Zhou et al. (2010), Bian & Tao (2011), Si et al. (2010), Zhang et al. (2009)), with applications including, e.g., human face recognition (Fu & Huang (2008)), speech and gait recognition (Tao et al. (2007)), etc.. The classical approaches of subspace selection in dimension reduction include algorithms like Principle Component Analysis (PCA, see Jolliffe (2002)) and Linear Discriminant Analysis (LDA, see Belhumeur et al. (1997), Tao et al. (2009)). They are looking for globally linear subspace models. Therefore, they fail to estimate the nonlinearity of the intrinsic data manifold, and ignore the local variation of the data (Saul & Roweis (2003), Strassen (1969)). Consequently, these globally linear models are often ineffective for search problems on large scale image data sets. To resolve this difficulty, nonlinear algorithms such as kernel algorithms (Ham et al. (2004)) and manifold learning algorithms (Belkin et al. (2006), Guan et al. (2011)) are proposed. However, even though these nonlinear methods significantly improve the recognition performance, they face a serious computational challenge dealing with large-scale data sets due to the complexity of matrix decomposition at the size of the number of training samples.

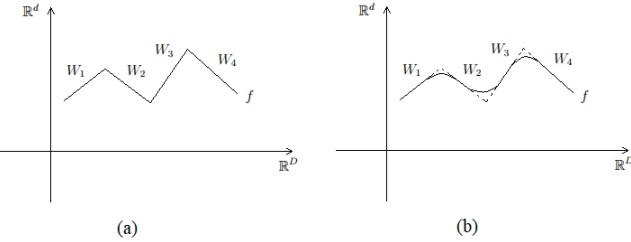

(a)          (b)

Figure 1: The idea of "smoothing" a piece-wise linear low-dimensional embedding model: (a) The piece-wise linear low-dimensional embedding model built from LPP; (b) The regularized low-dimensional embedding by taking Stiefel/Grassmann manifold center-of-mass among adjacent linear pieces.

Here we propose a simple method, Subspace Indexing Model with Interpolation (SIM-I), that produces from a given data set a piece-wise linear, locality-aware and globally nonlinear model of low-dimensional embedding. SIM-I is constructed via two steps: in the first step we build a piece-wise linear affinity-aware subspace model under a given partition of the data set; in the second step we interpolate between several adjacent linear subspace models constructed previously using the "center of mass" calculation on Stiefel and Grassmann manifolds (Edleman et al. (1999), Kaneko et al. (2013), Marrinan et al. (2014)). The interpolation step outputs a "smoothed" version (Figure 1) of the original piece-wise linear model, and can be regarded as a regularization process. Compared to previously mentioned subspace methods, SIM-I enjoys the following advantages: (1) it captures the *global nonlinearity* and thus the local fluctuations of the data set; (2) it is *computationally feasible* to large-scale data sets since it avoids the complexity in matrix decomposition at the size of the number of training samples; (3) it includes a *regularization step* via interpolating between several adjacent pieces of subspace models. Numerical experiments on PCA recovery task for SIFT data set and classification tasks via nearest-neighbor method for MNIST and CIFAR-10 data sets further validate the effectiveness of SIM-I.

## 2 PIECE-WISE LINEAR LOCALITY PRESERVING PROJECTION (LPP) MODEL

If an image data point $x \in \mathbb{R}^D$ is represented as a vector in a very high-dimensional space, then we want to find a low-dimensional embedding $y = f(x) \in \mathbb{R}^d$, $d \ll D$ such that the embedding function $f$ retains some meaningful properties of the original image data set, ideally close to its intrinsic dimension. If we restrict ourselves to linear maps of the form $y = W^T x \in \mathbb{R}^d$, where the $D \times d$ projection matrix $W = (w_{ij})_{1 \leq i \leq D, 1 \leq j \leq d}$ (assuming full rank), then such a procedure is called a locally linear low-dimensional embedding (see Roweis & Saulm (2000); Van Der Maaten et al. (2009)). The target is to search for a "good" projection matrix $W$, such that the projection $x \mapsto y = W^T x$ must preserve certain *locality* in the data set (this is called a *Locality Preserving Projection*, or LPP projection, see He & Niyogi (2003)). The locality is interpreted as a kind of intrinsic relative geometric relations between the data points in the original high-dimensional space, usually represented by the affinity matrix $S = (s_{ij})_{1 \leq i,j \leq n}$ (which is a symmetric matrix with non-negative terms). As an example, given unlabelled data points $x_1, ..., x_n \in \mathbb{R}^D$, we can take $s_{ij} = \exp\left(-\frac{\|x_i - x_j\|^2}{2\sigma^2}\right)$ when $\|x_i - x_j\| < \varepsilon$ and $s_{ij} = 0$ otherwise. Here $\sigma > 0$ and $\varepsilon > 0$ is a small threshold parameter, and $\|x_i - x_j\|$ is the Euclidean norm in $\mathbb{R}^D$. Based on the affinity matrix $S = (s_{ij})$, the search for the projection matrix $W$ can be formulated as the following optimization problem

$$\min_W \phi(W) = \frac{1}{2} \sum_{i,j=1}^n s_{ij} \|y_i - y_j\|^2 , \tag{1}$$

in which $y_i = W^T x_i$ and $y_j = W^T x_j$ and the norm $\|y_i - y_j\|$ is taken in the projected space $\mathbb{R}^d$. Usually when $\|x_i - x_j\|$ is large, the affinity $s_{ij}$ will be small, and vice versa. Thus (1) is seeking for the embedding matrix $W$ such that close pairs of image points $x_i$ and $x_j$ will be mapped to close pairs of embeddings $y_i = W^T x_i$ and $y_j = W^T x_j$, and vice versa. This helps to preserve the local geometry of the data set, a.k.a the locality. To solve (1), we introduce a weighted fully-connected graph $G$ where the vertex set consists of all data points $x_1, ..., x_n$ and the weight on the edge connecting $x_i$ and $x_j$ is given by $s_{ij} \geq 0$. Consider the diagonal matrix $\mathcal{D} = \text{diag}(\mathcal{D}_{11}, ..., \mathcal{D}_{nn})$ where $\mathcal{D}_{ii} = \sum_{j=1}^n s_{ij}$, and we then introduce the *graph Laplacian* $L = \mathcal{D} - S$. Then the minimization problem (1), together with the normalization constraint $\sum_{i=1}^n \mathcal{D}_{ii} y_i^2 = 1$, reduces to the following *generalized eigenvalue problem* (see He & Niyogi (2003))

$$X L X^T w = \lambda X \mathcal{D} X^T w , \tag{2}$$

where $X = [x_1, ..., x_n] \in \mathbb{R}^{D \times n}$.

Assume we have obtained an increasing family of eigenvalues $0 = \lambda_0 < \lambda_1 \leq ... \leq \lambda_{n-1}$. Let the corresponding eigenvectors be $w_0, w_1, ..., w_{n-1}$. Then the low-dimensional embedding matrix can

be taken as $W = [w_1, ..., w_d]$ (see He et al. (2005)). By choosing different affinity matrices $S = (s_{ij})$, the above LPP framework includes many commonly seen practical examples. For example, if the data $x_1, ..., x_n$ are not labelled, then we can take $s_{ij} = \dfrac{1}{n}$ and (2) produces the classical Principle Component Analysis (PCA). For labelled data forming subsets $\mathcal{X}_1, ..., \mathcal{X}_m$ with same labels in each subset, we can take $s_{ij} = \dfrac{1}{n_k}$ when $x_i, x_j \in \mathcal{X}_k$, and $s_{ij} = 0$ other-wise. Here $n_k$ is the cardinality of $\mathcal{X}_k$. This will produce Linear Discriminant Analysis (LDA). The detailed justifications of these connections can be found in He et al. (2005).

Given an input data set $\mathcal{X} = \{x_1, ..., x_n\}$ where each $x_i \in \mathbb{R}^D$, either labelled or unlabelled, we can apply a k-d tree (Bentley (1975), Wang et al. (2011)) based partition scheme to divide the whole data set $\mathcal{X}$ into non-overlapping subsets $C_1, ..., C_{2^h}$ where $h$ is the depth of the tree. Conventional subspace selection algorithms could be applied on the whole sample space before the whole space is partitioned and indexed. For example, we can first apply a PCA to $\mathcal{X}$, which selects the first $d$ bases $[a_1, ..., a_d]$ with largest variance. Based on these bases, the covariance information obtained from global PCA is utilized in the indexing as follows: (1) we project all sample points $x_1, ..., x_n$ onto the maximum variance basis $a_1$, then we find the median value $\mathfrak{m}_1$ of the projected samples, and split the whole collection of data along $a_1$ at $\mathfrak{m}_1$, i.e., split the current node into left and right children; (2) starting from level $i = 2$, for each left and right child, project the whole collection of data along the $i$-th maximum variance basis $a_i$, find the median value $\mathfrak{m}_i$, and split all the children at $\mathfrak{m}_i$; (3) increase the level from $i$ to $i + 1$ and repeat (2) until $i = h$ reaches the bottom of the tree. We collect all the $2^h$ children at level $i = h$ and obtain the disjoint subsets $C_1, ..., C_{2^h}$. Each subset $C_k$, $k = 1, 2, ..., 2^h$ consists of a family of input data in $\mathbb{R}^D$. Based on them, using the above LPP framework, for each $C_k$, a low-dimensional embedding matrix $W_k \in \mathbb{R}^{D \times d}$ can be constructed. In this way, over the whole data set $\mathcal{X}$, we have constructed a piece-wise linear low-dimensional embedding model $f(x) : \mathbb{R}^D \to \mathbb{R}^d$ (see Figure 1(a)) where $x \in \mathcal{X}$. This model is given by the linear embedding matrices $W_1, ..., W_{2^h} \in \mathbb{R}^{D \times d}$.

The above model construction can be regarded as a training process from the data set $\mathcal{X}$. For a given test data point $x \in \mathbb{R}^D$, not included in $\mathcal{X}$, we can find the closest subset $C_{k(x)}$ to it, by selecting the index $k = k(x) \in \{1, ..., 2^h\}$ with the smallest distance $\|x - m_{k(x)}\|$. Here $m_k$ is the mean among all data points in the subset $C_k$. With the subset $C_{k(x)}$ chosen, we map the test point $x \in \mathbb{R}^D$ to its low-dimensional embedding $f(x) = W_{k(x)}^T x \in \mathbb{R}^d$. Such a procedure extended the piece-wise linear embedding model $f(x) : \mathbb{R}^D \to \mathbb{R}^d$ to all testing data points in $\mathbb{R}^D$.

## 3 CALCULATING THE "CENTER OF MASS" ON STIEFEL AND GRASSMANN MANIFOLDS

*Subspace Indexing* (Wang et al. (2011)) provides a $d$-dimensional representation of the data set $\{x_1, ..., x_n\}$ by the subspace $\text{span}(w_1, ..., w_d) = \{W^T x, x \in \mathbb{R}^D\}$ generated from the linear embedding matrix $W \in \mathbb{R}^{D \times d}$. In this case, we are only interested in the column space of $W$, so we can assume that $w_0, w_1, ..., w_{n-1}$ is an orthonormal basis [1]. Such a matrix $W$ belongs to the *Stiefel manifold*, defined by

**Definition 1 (Stiefel manifold)** *The compact* Stiefel manifold $\text{St}(d, D)$ *is a submanifold of the Euclidean space* $\mathbb{R}^{D \times d}$ *such that*

$$\text{St}(d, D) = \{X \in \mathbb{R}^{D \times d} : X^T X = I_d\}. \tag{3}$$

As an example, if we are interested in signal recovery using low-dimensional PCA embedding, the projections we calculated from PCA analysis will be on Stiefel manifolds. However, for classification tasks, the exact distance information is less important than label information. In this case, two such Stiefel matrices $W_1$ and $W_2$ produce the same embedding if $W_1 = W_2 O_d$ for some $O_d \in O(d)$, where $O(d)$ is the group of orthogonal matrices in dimension $d$. In this case, the relevant embedding we obtained is a point on the *Grassmann manifold*, defined by

---

[1] If this is not the case, we can replace the matrix $[w_0 \ w_1 \ ... \ w_{n-1}]$ by the $Q$ matrix of the QR-decomposition of itself, without changing the corresponding subspace.

**Definition 2 (Grassmann manifold)** *The Grassmann manifold* $\mathrm{Gr}(d, D)$ *is defined to be the quotient manifold* $\mathrm{Gr}(d, D) = \mathrm{St}(d, D)/O(d)$. *A point on* $\mathrm{Gr}(d, D)$ *is defined by an equivalence class* $[W] = \{WO_d, O_d \in O(d)\}$ *where* $W \in \mathrm{St}(d, D)$.

Given a family of elements on the Stiefel or Grassmann manifold, the *center-of-mass* is defined as an element on the same manifold that minimizes the functional given by the weighted sum of square distances. To be precise, we have

**Definition 3 (Stiefel and Grassmann center-of-masses)** *Given a sequence of matrices* $W_1, ..., W_l \in \mathrm{St}(d, D)$ *and a sequence of weights* $w_1, ..., w_l > 0$, *the* Stiefel center-of-mass *with respect to the distance* $\mathrm{d}(W_1, W_2)$ *on* $\mathrm{St}(d, D)$ *is defined as a matrix* $W_c = W_c^{\mathrm{St}}(W_1, ..., W_l; w_1, ..., w_l) \in \mathrm{St}(d, D)$ *such that*

$$W_c = W_c^{\mathrm{St}}(W_1, ..., W_l; w_1, ..., w_l) \equiv \arg \min_{W \in \mathrm{St}(d,D)} \sum_{j=1}^{l} w_j \mathrm{d}^2(W, W_j). \tag{4}$$

*Similarly, if the corresponding equivalent classes are* $[W_1], ..., [W_l] \in \mathrm{Gr}(d, D)$, *then the* Grassmann center-of-mass *with respect to the distance* $\mathrm{d}([W_1], [W_2])$ *on* $\mathrm{Gr}(d, D)$ *is defined as the equivalence class* $[W_c]$, *where* $W_c = W_c^{\mathrm{Gr}}(W_1, ..., W_l; w_1, ..., w_l) \in \mathrm{St}(d, D)$ *is such that*

$$W_c = W_c^{\mathrm{Gr}}(W_1, ..., W_l; w_1, ..., w_l) \equiv \arg \min_{W \in \mathrm{St}(d,D)} \sum_{j=1}^{l} w_j \mathrm{d}^2([W], [W_j]). \tag{5}$$

The distances $\mathrm{d}(W_1, W_2)$ or $\mathrm{d}([W_1], [W_2])$ can be taken in different ways. For example, for $W_1, W_2 \in \mathrm{St}(d, D)$, one way is to consider $\mathrm{d}(W_1, W_2) = \mathrm{d}_F(W_1, W_2) = \|W_1 - W_2\|_F$, the matrix Frobenius norm of $W_1 - W_2$. One can also take a more intrinsic distance, such as the geodesic distance between $W_1$ and $W_2$ on the manifold $\mathrm{St}(d, D)$ with the metric given by embedded geometry (see Edleman et al. (1999)). For $[W_1], [W_2] \in \mathrm{Gr}(d, D)$, one way is to consider the projected Frobenius norm $\mathrm{d}([W_1], [W_2]) = \mathrm{d}_{pF}([W_1], [W_2]) = 2^{-1/2}\|W_1 W_1^T - W_2 W_2^T\|_F$. There are also many other choices, such as using the principle angles between the subspaces, chordal norms, or other types of Frobenius norms (see Edleman et al. (1999, Section 4.3)).

With respect to matrix Frobenius norm and projected Frobenius norm, the Stiefel and Grassmann center-of-masses can be calculated explicitly in the following theorems.

**Theorem 1 (Stiefel center-of-mass with respect to Frobenius norm)** *We consider the singular value decomposition of the matrix* $\sum_{j=1}^{l} w_j W_j = O_1 \Delta O_2$, *where* $O_1 \in O(D)$ *and* $O_2 \in O(d)$, $\Delta = \begin{pmatrix} diag(\lambda_1, ..., \lambda_d)_{d \times d} \\ 0_{(D-d) \times d} \end{pmatrix}$ *and* $\lambda_1 \geq ... \geq \lambda_d \geq 0$ *are the singular values. Then the Stiefel center-of-mass with respect to the distance given by Frobenius norm* $\mathrm{d}(W_1, W_2) = \|W_1 - W_2\|_F$ *is given by* $W_c = O_1 \Lambda O_2$ *where* $\Lambda = \begin{pmatrix} diag(1, ..., 1)_{d \times d} \\ 0_{(D-d) \times d} \end{pmatrix}$.

**Theorem 2 (Grassmann center-of-mass with respect to projected Frobenius norm)** *Set* $\Omega_j = \left(\sum_{j=1}^{l} w_j\right)^{-1} w_j$. *We consider the singular value decomposition of the symmetric matrix* $\sum_{j=1}^{l} \Omega_j W_j W_j^T = Q \Delta Q^T$ *where* $Q \in O(D)$ *and* $\Delta = diag(\sigma_1^2, ..., \sigma_D^2)$, $\sigma_1^2 \geq ... \geq \sigma_D^2 \geq 0$. *Then the Grassmann center-of-mass with respect to the distance given by projected Frobenius norm* $\mathrm{d}_{pF}([W_1], [W_2]) = 2^{-1/2}\|W_1 W_1^T - W_2 W_2^T\|_F$ *is the equivalence class* $[W_c]$ *determined by* $W_c = Q\Lambda$, *where* $\Lambda = \begin{pmatrix} diag(1, ..., 1)_{d \times d} \\ 0_{(D-d) \times d} \end{pmatrix}$.

## 4 SIM-I: INTERPOLATING THE LPP MODEL FAMILY

Recall that we have developed a piece-wise linear embedding model $f(x) : \mathbb{R}^D \to \mathbb{R}^d$ over the data set $\mathcal{X} = \{x_1, ..., x_n\}$. The embedding $f(x)$ corresponds to a family of subspace indexing models

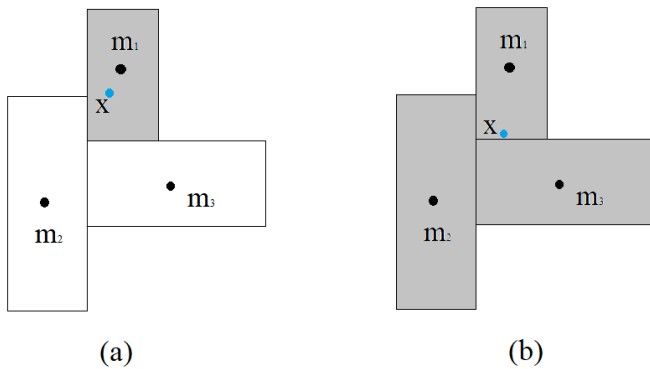

(a)    (b)

Figure 2: There are $I = 3$ nearby subsets $C_1, C_2, C_3$ with means $m_1, m_2, m_3$ in the training set. (a) Test point $x$ is apparently close to $m_1$, and thus the low-dimensional embedding $f(x)$ is taken as $f(x) = W_1^T x$, where $W_1$ is the LPP subspace based on $C_1$; (b) Test point $x$ has approximately the same distances to $m_1, m_2, m_3$, and thus the embedding $f(x)$ is taken as $f(x) = W_c^T x$, where $W_c$ is the Stiefel/Grassmann center-of-mass for the LPP subspace models $W_1, W_2, W_3$ based on $C_1, C_2, C_3$.

$W_1, ..., W_{2^h} \in \text{St}(d, D)$ (e.g. for PCA signal recovery tasks) or $[W_1], ..., [W_{2^h}] \in \text{Gr}(d, D)$ (e.g. for classification tasks). Each subspace model $W_k$ is built from LPP embedding using the subset $C_k \subset \mathcal{X}$ developed from k-d tree and $h$ is the depth tree. Given a test point $x \in \mathbb{R}^D$ that does not lie in $\mathcal{X}$, we can map it to the low-dimensional embedding $f(x) = W_{k(x)}^T x \in \mathbb{R}^d$. The index $k(x)$ corresponds to the subset $C_{k(x)}$ that lie closest to $x$. In practice, we can first compute the means $m_k$ over all the data points in the subset $C_k$ for each $k = 1, 2, ..., 2^h$ and sort the distances $\|x - m_k\|$ in ascending order $\|x - m_{k_1(x)}\| \leq ... \leq \|x - m_{k_{2^h}(x)}\|$, $\{k_1(x), ..., k_{2^h}(x)\} = \{1, ..., 2^h\}$. We then take $k(x) = k_1(x)$ to be the index $k$ corresponding to the shortest distance. This is effective when the test point $x$ lies significantly close to one of the subsets $C_{k(x)}$, see Figure 2(a).

---

**Algorithm 1** SIM-I: Subspace Indexing Model with Interpolation

---

1: **Input**: Data set $\mathcal{X} = \{x_1, ..., x_n \in \mathbb{R}^D\}$ and its corresponding affinity matrix $S = (s_{ij})_{1 \leq i,j \leq n}$; test point $x \in \mathbb{R}^D$; threshold ratio $r_{\text{thr}} > 1$; tree depth $h$; parameter $K > 0$

2: Using an initial PCA and a k-d tree based partition scheme, decompose the data set $\mathcal{X}$ into subsets $C_1, ..., C_{2^h}$, where $h$ is the depth of the tree

3: For each subset $C_k$, calculate its mean (center) $m_k \in \mathbb{R}^D$ and its LPP embedding matrix $W_k \in \text{St}(d, D)$ based on the affinity matrix $S$

4: Sort the distances $\|x - m_k\|$ in ascending order $\|x - m_{k_1(x)}\| \leq ... \leq \|x - m_{k_{2^h}(x)}\|$, $\{k_1(x), ..., k_{2^h}(x)\} = \{1, ..., 2^h\}$

5: Determine $I$, which is the first sub-index $i$ of $k_i(x)$ such that $\|x - m_{k_{I+1}(x)}\| > r_{\text{thr}} \|x - m_{k_1(x)}\|$

6: Set $j_i(x) = k_i(x)$ for $i = 1, 2, ..., I$ and obtain the embedding matrices $W_{j_1(x)}, ..., W_{j_I(x)} \in \text{St}(d, D)$ or their corresponding subspaces $[W_{j_1(x)}], ..., [W_{j_I(x)}] \in \text{Gr}(d, D)$, together with the weights $w_i = \exp(-K \|x - m_{j_i(x)}\|^2) > 0$ for $i = 1, ..., I$

7: Find a center-of-mass $W_c = W_c^{\text{St}}(W_{j_1(x)}, ..., W_{j_I(x)}; w_1, ..., w_I)$ (Stiefel case) or $[W_c] = [W_c^{\text{Gr}}(W_{j_1(x)}, ..., W_{j_I(x)}; w_1, ..., w_I)]$ (Grassmann case) according to Definition 3 and Theorems 1 and 2.

8: **Output**: The low-dimensional embedding $f(x) = W_c^T x \in \mathbb{R}^d$

---

However, for a general test point $x \in \mathbb{R}^D$, it might happen that this point lies at approximately the same distances to the centers of each of the several different subsets adjacent to $x$ (see Figure 2(b)).

In this case, we aim to *interpolate* between several subspace indexing models $W_{j_1(x)}, ..., W_{j_I(x)}$. To do this, we first find the subspace indexes $j_1(x), ..., j_I(x)$ from the first $I$ subsets $C_{j_1(x)}, ..., C_{j_I(x)}$ closest to $x$, i.e., $j_1(x) = k_1(x), ..., j_I(x) = k_I(x)$ given the sorted distances $\|x - m_k\|$ mentioned above. In practice, the number $I = I(x)$ is depending on $x$ and can be chosen in the following way: $I$ is the first sub-index $i$ of $k_i(x)$ such that $\|x - m_{k_{I+1}(x)}\| > r_{\text{thr}}\|x - m_{k_1(x)}\|$, where $r_{\text{thr}} > 1$ is a threshold ratio that can be tuned. We then pick the weights as $w_i = \exp(-K\|x - m_{j_i(x)}\|^2)$ for some $K > 0$ and $i = 1, 2, ..., I$. This is indicating that the closer $x$ is to $C_{j_i(x)}$, the heavier weights we assign to $W_{j_i(x)}$ in the interpolation process. Given the embedding matrices $W_{j_1(x)}, ..., W_{j_I(x)} \in \text{St}(d, D)$ or their corresponding subspaces $[W_{j_1(x)}], ..., [W_{j_I(x)}] \in \text{Gr}(d, D)$, together with the weights $w_1, ..., w_I > 0$, we find a center-of-mass $W_c = W_c^{\text{St}}(W_{j_1(x)}, ..., W_{j_I(x)}; w_1, ..., w_I)$ (Stiefel case) or $[W_c] = [W_c^{\text{Gr}}(W_{j_1(x)}, ..., W_{j_I(x)}; w_1, ..., w_I)]$ (Grassmann case) according to Definition 3 and Theorems 1 and 2. Finally, we map the test point $x$ to the low-dimensional embedding $f(x) = W_c^T x \in \mathbb{R}^d$. Notice that when $I = 1$, the interpolation procedure reduces to projecting $x$ using $W_{k_1(x)}$ calculated from LPP analysis on the closest subset only. In general, the whole interpolation procedure can be regarded as providing a regularized version of the piece-wise linear embedding we discussed in Section 2 (also see Figure 1). We summarize our interpolation method as the `SIM-I` Algorithm 1.

## 5 EXPERIMENTS

### 5.1 PCA RECOVERY FOR SIFT DATA SET

SIFT (Scale Invariant Feature Transform, see Lowe (2004)) data set is a data set that computes for each keypoint a real-valued descriptor, based on the content of the surrounding patch in terms of local intensity gradients. Given its remarkable performance, SIFT has been often used as starting point for the creation of other descriptors. The final SIFT descriptor consists of 128 elements. This means that each data point in SIFT data set has dimension $D = 128$, and we pick the embedding dimension $d = 16$. The original SIFT data set has 10068850 data samples, that form the data set `sift_sample`. We randomly collect $n_{\text{train}} = 200 \times 2^{13}$ elements from these sample points as our target data set $\mathcal{X} = \text{sift\_train} = \{x_1, ..., x_{n_{\text{train}}} \in \mathbb{R}^D\}$. We consider the recovery efficiency of PCA embedding of the SIFT data set. Let $x$ be a point in `sift_sample` and let $W$ be a Stiefel matrix in $\text{St}(16, 128)$. The projection of $x$ onto the 16-dimensional space is then denoted by $y = W^T x$. By recovery we meant to consider the point $\widehat{x} = (W^T)^- y$ where $(W^T)^-$ is the Moore-Penrose pseudo-inverse of $W^T$. The recovery efficiency can then be quantified by the recovery error $\|x - \widehat{x}\|$, where the Euclidean norm is computed in $\mathbb{R}^{128}$.

We pick $h = 13$ so that $2^h = 8192$. Then we decompose `sift_train` into 8192 subsets $C_1, ..., C_{8192}$ using k-d tree based partition. For each subset $C_k$, we calculate the mean $m_k \in \mathbb{R}^{128}$ and we obtain a PCA embedding matrix $W_k \in \text{St}(16, 128)$. We sort the distances $\|x - m_k\|, k = 1, 2, ..., 8192$ in ascending order so that $\|x - m_{k_1}\| \leq ... \leq \|x - m_{k_{8192}}\|$, where $\{k_1, ..., k_{8192}\} = \{1, ..., 8192\}$. Then we find among the subset means $m_k, k = 1, 2, ..., 8192$ the first $I$ nearest to $x$, with their indexes denoted by $k_i = k_i(x), i = 1, .., I$. The number $I = I(x)$ is depending on $x$ and is chosen in the following way: $I$ is the first sub-index $i$ of $k_i$ such that $\|x - m_{k_{I+1}}\| > r_{\text{thr}}\|x - m_{k_1}\|$, where $r_{\text{thr}} > 1$ is a threshold ratio. We pick $r_{\text{thr}} = 2$.

For test data set, we randomly pick from `sift_sample\sift_train` a subset of size $n_{\text{test}} = 500$, and we denote the data set as `sift_test`. For each test point $x \in$ `sift_test`, we consider sending it to the nearest subset $C_{k_1}$ and the corresponding Stiefel matrix is $W_{k_1} \in \text{St}(16, 128)$. We can then consider the recovery point $\widehat{x} = (W_{k_1}^T)^- W_{k_1}^T x$ and the benchmark recovery error Error_bm $= \|x - \widehat{x}\|$.

Consider the alternative recovery scheme using our method `SIM-I`. We calculate the weights $w_j = \exp(-K\|x - m_{k_j}\|^2)$ for $j = 1, ..., I$ and we choose the constant $K = 10^{-8}$. We then find the Stiefel center-of-mass $W_c = W_c^{\text{St}}(W_{k_1(x)}, ..., W_{k_I(x)}; w_1, ..., w_I)$ using Theorem 1 and taking the distance function to be the matrix Frobenius norm $\text{d}(W_1, W_2) = \|W_1 - W_2\|_F$. We consider the recovery point $\widehat{x}_c = ((W_c)^T)^- (W_c)^T x$ and the recovery error Error_c $= \|x - \widehat{x}_c\|$.

Over the test set `sift_test`, we find that for about $94.2\%$ test points, Error_c $<$ Error_bm, which implies that `SIM-I` improved the efficiency of recovery. The empirical average Error_c is

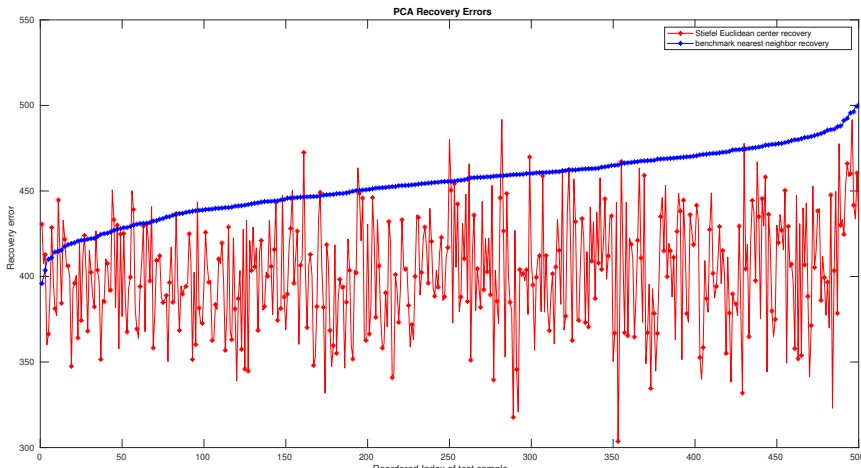

Figure 3: Comparison of the PCA Recovery Errors: Blue = benchmark case using closest subset PCA recovery, with the error sorted from low to high; Red = using SIM-I based on Stiefel center-of-mass and $\mathrm{d}(W_1, W_2) = \|W_1 - W_2\|_F$.

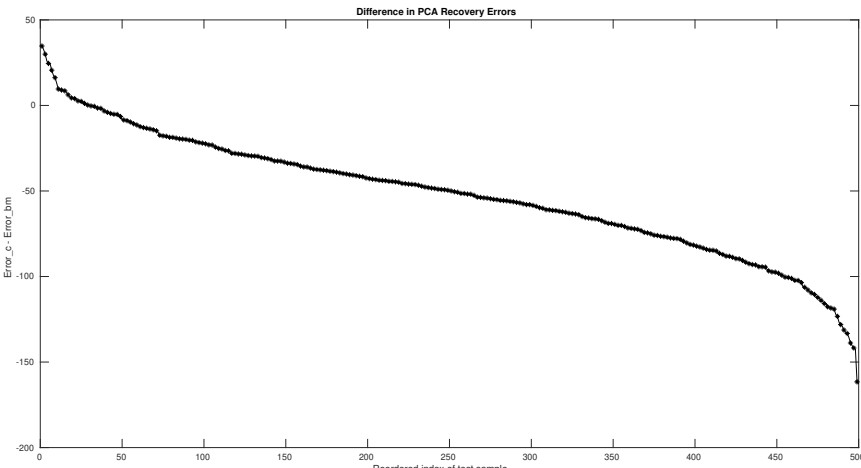

Figure 4: Differences Error_c − Error_bm in descending order.

402.089506, and the empirical average Error_bm is 454.452314. Figure 3 plots the Error_c and Error_bm (vertical axis) as functions of the test sample indexes. The red curve is for Error_c and blue curve is for Error_bm, where we have sorted Error_bm in ascending order and reordered the test indexes correspondingly. Figure 4 gives the differences Error_c − Error_bm in descending order. It can be apparently seen that Error_c − Error_bm < 0 for most of test samples.

## 5.2 NEAREST-NEIGHBOR CLASSIFICATION FOR MNIST AND CIFAR-10

Here we consider a labeled data set $\mathtt{data} = \{(x_i, y_i), i = 1, 2, ..., N\}$ where $x_i \in \mathbb{R}^D$ are the inputs and $y_i \in \mathbb{N}$ are the labels. We randomly select the training set $\mathtt{data\_train} = \{(x_i, Y_i), i = 1, 2, ..., n_{\mathrm{train}}\}$ and the test set $\mathtt{data\_test} = \{(a_i, b_i), i = 1, 2, ..., n_{\mathrm{test}}\}$, and we make them disjoint. We first project the training set onto a $\mathtt{kd\_PCA}$-dimensional subspace via a standard PCA.

Based on this initial embedding, using a kd-tree with height $h$, we divide `data_train` into $2^h$ clusters $C_k, k = 1, 2, ..., 2^h$. For each $C_k$, we find an LPP embedding matrix $W_k \in \mathrm{St}(\texttt{kd\_LPP}, D)$ by setting the affinity matrix to be $s_{ij} = \exp(-\|x_i - x_j\|^2)$ if $x_i, x_j \in C_k$ are in the same class and $s_{ij} = 0$ otherwise. Since we are having a classification problem, we can identify each $W_k$ by the subspace it spans, i.e., we consider $[W_k]$ which is the equivalence class of $W_k$ in $\mathrm{Gr}(\texttt{kd\_LPP}, D)$.

As before, we compute in $\mathbb{R}^D$ the means $m_k$ of each cluster $C_k$. For a test point $x \in$ `data_test`, we sort the distances $\|x - m_k\|, k = 1, 2, ..., 2^h$ in ascending order so that $\|x - m_{k_1}\| \leq ... \leq \|x - m_{k_{2^h}}\|$, where $\{k_1, ..., k_{2^h}\} = \{1, ..., 2^h\}$. Then we find among the cluster means $m_k, k = 1, 2, ..., 2^h$ the first $I$ nearest to $x$, with their indexes denoted by $k_i = k_i(x), i = 1, ..., I$. The number $I = I(x)$ is depending on $x$ and is chosen in the following way: $I$ is the first sub-index $i$ of $k_i$ such that $\|x - m_{k_{I+1}}\| > r_{\mathrm{thr}}\|x - m_{k_1}\|$, where $r_{\mathrm{thr}} > 1$ is a threshold ratio. The parameter $r_{\mathrm{thr}}$ can be treated as a hyper-parameter that we can tune here.

For baseline method, we do a nearest-neighbor classification for $x$ on a low-dimensional embedding of the cluster $C_{k_1}$, and we pick the number of nearest-neighbors to be `knn` $\geq 1$. Indeed we project $x$ and all training data points in $C_{k_1}$ using $W_{k_1}$, and perform nearest-neighbor classification on the resulting projection.

For our method `SIM-I`, we take the union $C_{k_1} \cup ... \cup C_{k_I}$. Recall that each $C_{k_i}$ corresponds to an LPP embedding projection matrix $W_{k_i}$. We set the weights $w_i = \exp(-K\|x - m_{k_i}\|^2)$, and we pick $K = 10^{-8}$. We compute a center-of-mass $W_c$ of the projection matrices $W_{k_i}$ with weights $w_i, i = 1, 2, ..., I$ using the Grassmann center-of-mass method, where the distance is taken as the projected Frobenius norm of Grassmann matrices, i.e., $\mathrm{d}(W_1, W_2) = 2^{-1/2}\|W_1 W_1^T - W_2 W_2^T\|_F$ and $\| \bullet \|_F$ is the matrix Frobenius norm. We obtain $W_c = W_c^{\mathrm{Gr}}(W_{k_1(x)}, ..., W_{k_I(x)}; w_1, ..., w_I)$ and we project $x$ and all training data points in the union $C_{k_1} \cup ... \cup C_{k_I}$ using $W_c$. We then perform a nearest-neighbor classification for $x$ on this low-dimensional embedding with the number of nearest-neighbors being equal to `knn`.

Table 1: Nearest-neighbor classification success rates.

| data set | tree height | $r_{\mathrm{thr}}$ | knn | baseline | SIM-I with Grassmann center |
|----------|-------------|--------------------|-----|----------|-----------------------------|
| MNIST | 8 | 1.2 | 1 | 93.58% | 96.55% |
| MNIST | 8 | 1.2 | 75 | 87.52% | 94.21% |
| CIFAR-10 | 8 | 1.05 | 1 | 29.49% | 33.04% |
| CIFAR-10 | 8 | 1.1 | 75 | 30.25% | 32.74% |

We have been experimenting on 2 different data sets, for both of them we pick `kd_PCA = 128`, `kd_LPP = 100`: (1) The MNIST data set, with $N = 70000, n_{\mathrm{train}} = 60000, n_{\mathrm{test}} = 10000$ data points and $D = 784$; (2) The CIFAR-10 data set, with $N = 60000, n_{\mathrm{train}} = 50000, n_{\mathrm{test}} = 10000$ data points and $D = 3072$. Table 1 shows the results, where the last 2 columns are the nearest-neighbor classification success rates for baseline method, and for `SIM-I` using Grassmann center-of-mass and the rows are for different experiments. The first 4 columns are for data sets, kd-tree height, threshold value $r_{\mathrm{thr}}$, the number of nearest-neighbors `knn`, respectively. We slightly tuned $r_{\mathrm{thr}}$ to reach best performances. Clearly, `SIM-I` has its advantage.

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

## A  PROOF OF THEOREM 1

Let $D \geq d \geq 1$. Recall $\mathrm{St}(d, D)$ stands for the Stiefel manifold, that is, each matrix in $\mathrm{St}(d, D)$ is a $D$ by $d$ matrix with columns being orthogonally normal. For any matrix $M$, recall that $\|M\|_F^2 = [\mathrm{tr}(M^T M)]^{1/2}$ is the Frobenius norm of $M$.

**Theorem 1 (Stiefel center-of-mass with respect to Frobenius norm)** *We consider the singular value decomposition of the matrix* $\sum_{j=1}^{l} w_j W_j = O_1 \Delta O_2$*, where* $O_1 \in O(D)$ *and* $O_2 \in O(d)$*,* $\Delta = \begin{pmatrix} diag(\lambda_1, ..., \lambda_d)_{d \times d} \\ 0_{(D-d) \times d} \end{pmatrix}$ *and* $\lambda_1 \geq ... \geq \lambda_d \geq 0$ *are the singular values. Then the Stiefel center-of-mass with respect to the distance given by Frobenius norm* $\mathrm{d}(W_1, W_2) = \|W_1 - W_2\|_F$ *is given by* $W_c = O_1 \Lambda O_2$ *where* $\Lambda = \begin{pmatrix} diag(1, ..., 1)_{d \times d} \\ 0_{(D-d) \times d} \end{pmatrix}$*.*

**Proof.** For $W \in \mathrm{St}(d, D)$ we define $f(W) = \sum_{j=1}^{l} w_j \|W - W_j\|_F^2$, and we are looking for a minimizer $W_c$ of $f$ on $\mathrm{St}(d, D)$. Write

$$
\begin{aligned}
\|W - W_j\|_F^2 &= \mathrm{tr}[(W - W_j)^T (W - W_j)] \\
&= \mathrm{tr}(W^T W) + \mathrm{tr}(W_j^T W_j) - 2\mathrm{tr}(W^T W_j) \\
&= 2p - 2\,\mathrm{tr}(W^T W_j).
\end{aligned}
$$

It follows that

$$
\begin{aligned}
f(W) &= 2p \sum_{j=1}^{l} w_j - 2 \sum_{j=1}^{l} w_j \, \mathrm{tr}(W^T W_j) \\
&= 2p \sum_{j=1}^{l} w_j - 2\,\mathrm{tr}(W^T B)
\end{aligned}
$$

where $B := \sum_{j=1}^{l} w_j W_j$. By singular value decomposition, there are $D \times D$ orthogonal matrix $O_1$ and $d \times d$ orthogonal matrix $O_2$ such that $B = O_1 \Delta O_2$, where $\Delta := \begin{pmatrix} diag(\lambda_1, ..., \lambda_d)_{d \times d} \\ 0_{(D-d) \times d} \end{pmatrix}$ , and $\lambda_1 \geq \cdots \geq \lambda_d \geq 0$ are the singular values of $B$. It follows that $\mathrm{tr}(W^T B) = \mathrm{tr}(\Delta O_2 W^T O_1)$. Observe that $(O_2 W^T O_1)^T \in \mathrm{St}(d, D)$. Write $O_2 W^T O_1 = (c_{ij})_{d \times D}$. Then $\sum_{j=1}^{D} c_{ij}^2 = 1$ for each $i$. Hence

$$
\mathrm{tr}(W^T B) = \sum_{i=1}^{d} \lambda_i c_{ii} \leq \sum_{i=1}^{d} \lambda_i \ ,
$$

with equality holds when $O_2 W^T O_1 = (I_d, O)$ where $O$ is the $d \times (D - d)$ matrix with all entries being zero. This says that $W_c = O_1 \Lambda O_2$ where $\Lambda = \begin{pmatrix} diag(1, ..., 1)_{d \times d} \\ 0_{(D-d) \times d} \end{pmatrix}$ is the maximizer of $\mathrm{tr}(W^T B)$. The so obtained $W_c$ serves as the minimizer of $f(W)$ on $\mathrm{St}(d, D)$ and is thus the center-of-mass. $\qquad\square$

## B  PROOF OF THEOREM 2

Recall that $\mathrm{Gr}(d, D) = \mathrm{St}(d, D)/O(d)$. Every point $W$ on $\mathrm{St}(d, D)$ will correspond to an equivalence class $[W] = \{W O_d : O_d \in O(d)\}$ which is a point on $\mathrm{Gr}(d, D)$. To represent points on $\mathrm{Gr}(d, D)$ using matrices, we notice that every point on $\mathrm{Gr}(d, D)$ corresponds to a unique choice of matrix $WW^T$ where $W \in \mathrm{St}(d, D)$. In this way, we can define the projected Frobenius distance between two classes $[W_1]$ and $[W_2]$ in $\mathrm{Gr}(d, D)$ as $\mathrm{d}_{pF}^2([W_1], [W_2]) = 2^{-1/2}\|W_1 W_1^T - W_2 W_2^T\|_F$.

**Theorem 2 (Grassmann center-of-mass with respect to projected Frobenius norm)** *Set* $\Omega_j = \left(\sum_{j=1}^{l} w_j\right)^{-1} w_j$. *We consider the singular value decomposition of the symmetric matrix* $\sum_{j=1}^{l} \Omega_j W_j W_j^T = Q\Delta Q^T$ *where* $Q \in O(D)$ *and* $\Delta = diag(\sigma_1^2, ..., \sigma_D^2)$, $\sigma_1^2 \geq ... \geq \sigma_D^2 \geq 0$. *Then the Grassmann center-of-mass with respect to the distance given by projected Frobenius norm* $\mathrm{d}_{pF}([W_1], [W_2]) = 2^{-1/2}\|W_1 W_1^T - W_2 W_2^T\|_F$ *is the equivalence class* $[W_c]$ *determined by* $W_c = Q\Lambda$, *where* $\Lambda = \begin{pmatrix} diag(1, ..., 1)_{d \times d} \\ 0_{(D-d) \times d} \end{pmatrix}$.

**Proof.** Set $M = WW^T$ and $M_j = W_j W_j^T$, then we are looking for a minimizer $W_c \in \mathrm{St}(d, D)$ of $f(W) \equiv \sum_{j=1}^{l} w_j \mathrm{d}_{pF}^2([W], [W_j]) = 2^{-1} \sum_{j=1}^{l} w_j \|M - M_j\|_F^2$. For two $D \times D$ matrices $M_1, M_2$ we define $\langle M_1, M_2 \rangle = \mathrm{tr}(M_1^T M_2)$. It is easy to verify that $\langle, \rangle$ is an inner product and $\|M\|_F^2 = \langle M, M \rangle$. Thus we write

$$
\begin{aligned}
2f(W) &= \sum_{j=1}^{l} w_j \|M - M_j\|_F^2 \\
&= \sum_{j=1}^{l} w_j \langle M, M \rangle - 2 \sum_{j=1}^{l} w_j \langle M, M_j \rangle + \sum_{j=1}^{l} w_j \langle M_j, M_j \rangle \\
&= \sum_{j=1}^{l} w_j \left[ \langle M, M \rangle - \langle M, 2 \sum_{j=1}^{l} \Omega_j M_j \rangle \right] + \sum_{j=1}^{l} w_j \langle M_j, M_j \rangle \\
&= \sum_{j=1}^{l} w_j \langle M - \sum_{j=1}^{l} \Omega_j M_j, M - \sum_{j=1}^{l} \Omega_j M_j \rangle - \sum_{j=1}^{l} w_j \left\| \sum_{j=1}^{l} \Omega_j M_j \right\|_F^2 + \sum_{j=1}^{l} w_j \|M_j\|_F^2 \\
&= \sum_{j=1}^{l} w_j \left\| M - \sum_{j=1}^{l} \Omega_j M_j \right\|_F^2 - \sum_{j=1}^{m} w_j \left\| \sum_{j=1}^{l} \Omega_j M_j \right\|_F^2 + \sum_{j=1}^{l} w_j \|M_j\|_F^2 .
\end{aligned}
$$

So the minimum is taken when $\left\| M - \sum_{j=1}^{l} \Omega_j M_j \right\|_F^2$ is minimized. Since each $W_j W_j^T$ is a symmetric matrix, we consider the SVD decomposition $\sum_{j=1}^{l} \Omega_j W_j W_j^T = Q\Delta Q^T$ where $Q \in O(D)$ and $\Delta = \mathrm{diag}(\sigma_1^2, ..., \sigma_D^2), \sigma_1^2 \geq \sigma_2^2 \geq ... \geq \sigma_D^2 \geq 0$. Thus it suffices to set $W_c = \arg\min_{W \in \mathrm{St}(d,D)} \|WW^T - Q\Delta Q^T\|_F^2$. Since $W \in \mathrm{St}(d, D)$, we have $WW^T = PVP^T$ where $P$ is an orthogonal matrix of size $D \times D$ and $V = \mathrm{diag}(1, 1, .., 1, 0, 0, ..., 0)$ is an $D \times D$ matrix with $\mathrm{rank}(V) = d$. Moreover, $P$ can be chosen as $P = (W \ Z)$ where $Z$ is an $D \times (D - d)$ matrix. So

$$
\min_{W \in \mathrm{St}(d,D)} \|WW^T - QVQ^T\|_F^2 = \min_{P \in O(D)} \|PVP^T - Q\Delta Q^T\|_F^2 .
$$

Let the orthogonal matrix $O = Q^{-1}P$. Then we further have

$$
\begin{aligned}
\min_{P \in O(n)} \|PVP^T - Q\Delta Q^T\|_F^2 &= \min_{O \in O(n)} \|Q(OVO^T - \Delta)Q^T\|_F^2 \\
&= \min_{O \in O(n)} \mathrm{tr}(Q(OVO^T - \Delta)^2 Q^T) \\
&= \min_{O \in O(n)} \mathrm{tr}(OVO^T - \Delta)^2 .
\end{aligned}
$$

We then show that $\min_{O \in O(n)} \mathrm{tr}(OVO^T - \Delta)^2$ is achieved at the orthogonal matrix

$$
O^* = \begin{pmatrix} O_{p \times p} & 0 \\ 0 & O_{(n-p) \times (n-p)} \end{pmatrix}
$$

for any $O_{p \times p} \in O(p)$ and any $O_{(n-p) \times (n-p)} \in O(n-p)$. Indeed, we have $\text{tr}(OVO^T - \Delta)^2 = \text{tr}(OVO^T) + \text{tr}(\Delta^2) - 2\text{tr}(\Delta OVO^T)$, so that we only have to maximize $\text{tr}(\Delta OVO^T) = \sum_{i=1}^{D} \sigma_i^2 c_{ii}$ where $OVO^T = (c_{ij})_{D \times D}$. Let $O = (v_{ij})_{D \times D}$, then it is easy to calculate that $c_{ii} = \sum_{j=1}^{d} v_{ij}^2$.

Moreover since $\sum_{j=1}^{D} v_{ij}^2 = 1$ for all $i$ and $\sum_{i=1}^{D} v_{ij}^2 = 1$ for all $j$, we know that $0 \leq c_{ii} \leq 1$ and $\sum_{i=1}^{D} c_{ii} = d$. Thus from $\sigma_1^2 \geq ... \geq \sigma_D^2 \geq 0$ we see $\text{tr}(\Delta OVO^T) = \sum_{i=1}^{D} \sigma_i^2 c_{ii} \leq \sum_{i=1}^{d} \sigma_i^2$ with equality if and only if $c_{ii} = 1$ for $i = 1, ..., d$ and $c_{ii} = 0$ for $i = d+1, ..., D$. This gives that, for $O = (v_{ij})_{D \times D}$, we have $\sum_{j=1}^{d} v_{ij}^2 = 1$ when $i = 1, ..., d$ and $\sum_{j=d+1}^{D} v_{ij}^2 = 1$ when $i = d+1, ..., D$.

Thus the minimum of $\min_{O \in O(n)} \text{tr}(OVO^T - \Delta)^2$ is achieved at $O^* = \begin{pmatrix} O_{d \times d} & 0 \\ 0 & O_{(D-d) \times (D-d)} \end{pmatrix}$ for any $O_{d \times d} \in O(d)$ and any $O_{(D-d) \times (D-d)} \in O(D-d)$. We can pick $O_{d \times d} = \text{diag}(1, ..., 1)_{d \times d}$, and since $P^* = QO^* = (W_c \ Z)$ where $Z$ is an $D \times (D-d)$ matrix, we get the conclusion of the Theorem about $W_c$. $\qquad \square$

