# OpenReview forum: "Effective Subspace Indexing via Interpolation on Stiefel and Grassmann manifolds"
_ICLR.cc/2021/Conference — Reject_

### Official Review · AnonReviewer3 · 2020-10-25
**This proposed subspace index model with interpolation method seems reasonable but unclear why we have to do this, experimental results are reasonable but not strong enough.**

**Rating:** 5
**Confidence:** 4

**Review:**

### Summary:

This submission proposed a subspace index method with the additional interpolation (smoothing) step, i.e., SIM-I, first to build a piecewise linear affinity-aware subspace model under a given partition of the data set; next is to interpolate between several local linear subspace models by using the “center of mass” on Grassmann manifolds.

Experimental results on SIFT data, MNIST, and CIFAR-10 are included to support the effectiveness of SIM-I, including some comparisons with baseline methods.

### Strengths & Originality:

Subspace index plus interpolation is a reasonable idea, and view the interpolation step as some sort of regularization is interesting.

Computation complexity should be one of the advantage of this work, compared with many nonlinear dimension reduction methods need to solve an eigen-decomposition problem for a N-by-N matrix (N as number of samples). However, seems no clear summary of the total complexity for SIM-I in this submission.

Supplementary materiel is provided with the source code in MATLAB, this is good to see.

### Weakness:

The proposed method share similar framework with "Subspaces Indexing Model on Grassmann Manifold" (Ref [1], SIM-GM). We can see the differences by read through both papers, but, seems there is no clear summary for major contributions made against SIM-GM [1] by this submission.

The interpolation step in SIM-I, i.e., step 5-8 in Algorithm 1, can be viewed as some sort of "smoothing" from a number of subspace. Here the proposed "center-of-mass" on Grassmann manifolds is reasonable but not clear why, i,e., will be nice to see discussion or some theoretical analysis here. For example, there is a long history of local space alignment & smoothing works related to dimension reduction & manifold learning, e.g., one of the pointer work is LTSA (Ref. [4]) .

Experiential results on (1) reconstruction from embedding space (2) NN classification are reasonable, but there is no comparison with alternative methods, as only baseline method such as PCA is included in Section 5.1. For example, it will be informative to see the comparison with SIM-GM.

K-d tree is not quite effective in high dimensional space, and this is probably why the 1st step in both the proposed SIM-I and SIG-GM [1] is to use global linear PCA for dimension reduction. This seems fine, still raising some concerns for using PCA as the foundation step for non-linear low-dimensional embedding methods.

### Reference:

Related works from non-linear dimension reduction and manifold learning are discussed in this submission. Also, it seems GPCA (Ref. [5]) is related to the concept of subspace index model, so put it here.
1. Xinchao Wang, Zhu Li, and Dacheng Tao. Subspace Indexing on Grassmann Manifold for Large Scale Multimedia Retrieval. IEEE Trans on Image Processing, 2011
2. Xiaofei He and Partha Niyogi. Locality Preserving Projections. Advances in Neural Information Processing Systems (NIPS), 2003
3. Mikhai Belkin and Partha Niyogi. Laplacian Eigenmaps for Dimensionality Reduction and Data Representation, Neural Computation, 2003
4. Zhengyue Zhang and Hongyuan Zha. Principal Manifolds and Nonlinear Dimension Reduction via Local Tangent Space Alignment. SIAM Journal on Scientific Computing, 2005
5. Rene Vidal, Yi Ma, and Shankar Sastry. Generalized principal component analysis (GPCA). CVPR 2003

---

### Official Review · AnonReviewer4 · 2020-10-27
**Experiments are not enough, and the novelty is limited**

**Rating:** 4
**Confidence:** 3

**Review:**

Summary:
This paper proposes a subspace indexing model with interpolation (SIM-I) for dimension reduction. To capture the global nonlinearity and the local variation of data, SIM-I split the global space into a collection of local disjoint partitions by a kd-tree style scheme and build a subspace indexing model (e.g., LPP) for each local partition. The main contribution of this paper is to propose an interpolation technique to combine several subspace indexing models to determine a more promising model for low-dimensional embedding. Two experiments on PCA recovery and nearest neighbour classification validate the effectiveness of SIM-I.

Pros:
1. The authors introduce a new way to compute the centre-of-masses of Stiefel and Grassmann with respect to Frobenius norm and projected Frobenius norm with the theoretical analysis.
2. The idea of using interpolation to get a new model for the query locating at the boundary is interesting. Compared to [1], this idea seems simpler and more effective since it considers all of the close local partitions with a distance threshold.
3. The paper is clear to read. The authors present their algorithm step-by-step in detail. It is easy to reproduce the algorithms with the guideline.

Cons:
1. The key concern of this paper is the lack of rigorous experiments for the evaluation of SIM-I. More details can be found in minor comments.
2. The novelty is limited. The LPP model [4][5] and the kd-tree partitioning scheme [1] are not proposed by the authors. To make a clear difference between former work and the current work, I suggest the authors add a preliminary section to review these methods.
3. The discussion of related work is missing, and the references may be out-of-date since no reference is included in the recent five years.
4. As the authors claim their method can handle large scale data set, space and time complexity analysis should be included for an illustration. And it will be better to add a table to compare with other SOTA methods.

Minor Comments or Typos:
1. The benchmark method is too weak, which is only a special case of SIM-I. Despite the authors claimed there exists drawbacks of globally linear subspace model (e.g., PCA, LDA, and LPP) and nonlinear methods such as the kernel methods and manifold learning, they did not choose any of it as a competitor for further validation. This is very limited. At least, the authors should compare [1] with SIM-I since both of them use kd-tree for partitioning and belong to globally nonlinear methods.
2. The authors claim SIM-I is computationally feasible to large-scale data sets. However, this paper lacks experiments for validation. Compared to SOTA methods, how fast it can be? How large scale data it can handle in a reasonable time?
3.  Since the authors introduce a new way to compute the centre-of-masses of Stiefel and Grassmann, can the authors add an ablation study to verify the efficiency and effectiveness of this way compared with the methods [2][3] they mentioned in the section of introduction?
4. Moreover, the study of parameters is necessary. Is the model sensitive to tree depth h? threshold ratio r_{thr}? The parameter K for the weight?
5. Figure 1 is not clear. And the “LPP” has first appeared without illustration.
6. Consider the last two lines in Page 2,
\lambda_0 < \lambda_1 \leq … \leq \lambda_{n-1}  -->  \lambda_0 < \lambda_1 \leq … \leq \lambda_{d-1}
w_0, w_1,…, w _{n-1}  -->  w_0, w_1,…, w _{d-1}, and hence consider the first line of Page 3,
W=[w_1,w_2,…,w_d]  -->  W=[w_0, w_1,…,w_{d-1}]
7. Consider the first paragraph of Section 3 and the footnote 1 in Page 3,
w_0,w_1,…,w_{n-1}  -->  w_0, w_1, …, w_{d-1},
[w_0 w_1 … w_{n-1}]  --> [w_0 w_1 … w_{d-1}]
8. I suggest the authors add download link or citation for the three data sets used in the experiments, especially for the sift data set they used for the reproductivity purpose.
9. For the PCA recovery experiments, since the authors measure the recovery error, it would be better to call recovery effectiveness instead of recovery efficiency.

Reference

[1] Wang, Xinchao, Zhu Li, and Dacheng Tao. "Subspaces indexing model on Grassmann manifold for image search." IEEE Transactions on Image Processing 20, no. 9 (2011): 2627-2635.

[2] Kaneko, Tetsuya, Simone Fiori, and Toshihisa Tanaka. "Empirical arithmetic averaging over the compact Stiefel manifold." IEEE Transactions on Signal Processing 61, no. 4 (2012): 883-894.

[3] Marrinan, Tim, J. Ross Beveridge, Bruce Draper, Michael Kirby, and Chris Peterson. "Finding the subspace mean or median to fit your need." In Proceedings of the IEEE Conference on Computer Vision and Pattern Recognition, pp. 1082-1089. 2014.

[4] He, Xiaofei, and Partha Niyogi. "Locality preserving projections." In Advances in neural information processing systems, pp. 153-160. 2004.

[5] He, Xiaofei, Shuicheng Yan, Yuxiao Hu, Partha Niyogi, and Hong-Jiang Zhang. "Face recognition using laplacianfaces." IEEE transactions on pattern analysis and machine intelligence 27, no. 3 (2005): 328-340.


===============================================================================================

Update:
Since the authors did not give any feedback for the reviews, I retain my original decision. Thank you.

---

### Official Review · AnonReviewer1 · 2020-10-28
**A method for low-dimensional embedding based on interpolating between local subspaces.**

**Rating:** 3
**Confidence:** 2

**Review:**

This paper presents an approach to low-dimensional embedding of data, with an emphasis on image datasets. The algorithm proceeds by first finding several local subspaces that fit the data and then tying these subspaces together using the center of mass calculation on the Stiefel/Grassmann manifold. The method appears to be computationally efficient, since it can be applied to fairly large image datasets, though no formal analysis is given. The main drawback of the paper is that the empirical results compare only to PCA. Several more modern methods for low-dimensional embeddings exist (e.g., T-SNE, UMAP), and it would be appropriate to compare to these. UMAP also allows for the incorporation of labeled data.

Aside from this, the paper is quite difficult to follow due to the current organizational structure. A final minor comment is that the figure text is too small to be reasonably legible.

---

### Official Review · AnonReviewer2 · 2020-10-29
**not yet convincing enough for publication**

**Rating:** 4
**Confidence:** 5

**Review:**

The authors propose an approach towards computing  globally non-linear, low-dimensional embeddings of high dimensional data. In particular, they consider a sub-space indexing model with interpolation (called SIM-I) which consists of two steps. First, a locally linear model of the data is generated (using kD-trees and PCA) and second, averaging operations on the respective projection matrices are carried out. These are computed based on center of mass computations on Stiefel or Grasammanian manifolds. Anectdotal experimental evidence shows that this approach outperforms simple baseline techniques.

While non-linear dimensionality reduction is certainly of considerable practical interest, the work reported here is not yet convincing enough to merit publication.

First of all, there is a very rich literature on "local PCA" which is not acknowledged and the practical performance of corresponsing earlier methods is not considered / reported. Second of all, the authors list various previously popular techniques such as locally linear embeddings or Laplcaian eigenmaps but again do not consider those in their practical evaluation. As a consequence, it is difficult to assess the merits of the proposed approach over well known baselines. Third of all, while generally interesting, the proposed approach constitutes but a mash-up of existing ideas and can be considered  ad-hoc and a bit outdated. A clear cut motivation or provable guarantees of the proposed method are missing. Finally, the authors critize the fact that previous non-linear methods "face a serious computational challenge dealing with large-scale data sets due to the complexity" but do not provide evidence for a superior runtime behavior of their approach. Indeed, the experiments considered here, also only consider data sets of moderate sizes when compared to modern standards.

---

### Decision · Program_Chairs · 2021-01-07
**Final Decision**

**Decision:**

Reject

**Comment:**

Thanks for your submission to ICLR.

This paper proposes a subspace indexing model for low-dimensional embedding.  The reviewers were all generally in agreement that the paper is not ready for publication.  In particular, they felt that the paper had several key weaknesses:

-Relevant literature is not discussed
-Relevant methods are not evaluated against in the experiments
-Experiments on the whole were limited and not sufficiently convincing
-The novelty of the paper is not very high

Please consider the reviewer comments carefully when preparing a future version of your paper.